# High-Speed Maneuvering Target Inverse Synthetic Aperture Radar Imaging and Motion Parameter Estimation Based on Fast Spare Bayesian Learning and Minimum Entropy

Shuangzhi Xia [1], Yuanyuan Wang [2,*], Juan Zhang [2] and Fengzhou Dai [2]

[1] The 54th Research Institute of China Electronics Technology Group Corporation, Shijiazhuang 050081, China; 20021210952@stu.xidian.edu.cn

[2] National Key Laboratory of Radar Signal Processing, Xidian University, Xi'an 710071, China; jzhang@xidian.edu.cn (J.Z.); fzdai@xidian.edu.cn (F.D.)

* Correspondence: yywang_96@stu.xidian.edu.cn

**Abstract:** High-speed maneuvering target inverse synthetic aperture radar (ISAR) imaging is always a hot topic in signal processing. High-speed maneuvering targets often have high-order maneuvering characteristics, such as translational and rotational characteristics, which destroy the signal structure of stationary targets and make basic imaging processing methods such as range-Doppler (RD) algorithm no longer suitable. In this paper, a high-resolution imaging method for high-speed maneuvering targets is proposed, which uses the fast sparse Bayesian learning (SBL) algorithm and the minimum entropy algorithm for ISAR high-resolution imaging and motion parameter estimation, respectively. Because SBL makes full use of the characteristics of the target and the environment, it can obtain an ISAR high-resolution image of the maneuvering target. However, the high computational complexity caused by matrix inversion and some matrix operations in SBL iteration limit the practical application of SBL in ISAR imaging. In view of the special structure of the matrix required to be inverted, we propose a fast SBL algorithm, which uses a new decomposition method to obtain the decomposition formula of the inverse matrix. Based on the decomposition factors, the multiplication operation involving the inverse matrix can be quickly calculated using fast Fourier transform (FFT), which greatly improves the computational efficiency. Image entropy represents the sharpness and focusing degree of an image, and so the minimum entropy algorithm can estimate the motion parameters of maneuvering targets more accurately. We combine the minimum entropy algorithm with the fast SBL algorithm to realize phase error correction and high-resolution imaging, which has better noise sensitivity and can obtain the best focusing degree image. Finally, simulation results prove the effectiveness of this algorithm.

**Keywords:** high-speed maneuvering target high-resolution imaging; fast SBL algorithm; minimum entropy algorithm; phase error correction

## 1. Introduction

Since radar imaging is not restricted by environmental and time factors, it has the characteristics of all-weather, all-sky time, long-range action, and high resolution; therefore, more and more attention has been paid to developments in the field of remote sensing detection [1–6]. High-speed maneuvering targets, such as satellites, missiles and drones, and other space objects, often have large radar radial velocity and acceleration [7]. The high-speed maneuvering characteristics of targets broaden the one-dimensional-range image of the broadband imaging radar, and the combined action of high-order motion parameters such as velocity and acceleration makes it difficult to accurately compensate for the one-dimensional-range image, resulting in a fuzzy two-dimensional image. As a result, accurate detection, tracking, and imaging are not possible [8].

Inverse synthetic aperture radar (ISAR) imaging technology is aimed at non-cooperative targets, for which motion is uncertain and complex, and can obtain a two-dimensional image of the target [8,9]. The key condition of ISAR imaging is the rotation of the target relative to the radar. Two important aspects of ISAR imaging are high-resolution imaging algorithms and motion compensation. In the existing ISAR imaging algorithms, the traditional range-Doppler (RD) algorithm is widely used due to its small computation amount. But, RD can obtain well-focused images for stationary moving targets, for maneuvering targets, or for complex moving targets; after motion compensation, high-order phase signals brought by high-order motion parameters still exist in their echo data, and so the image obtained by RD is defocused, and the algorithm is invalid [10]. It is well known that the electromagnetic scattering characteristic of the radar target is in the high frequency region, and the echo signal of the target can be characterized by a few important scattering centers [11]. The sparse characteristics of radar target echoes are consistent with the requirement of the compressive sensing (CS) theory [11,12] on sparsity, and CS technology can reduce the sampling rate of radar systems. Therefore, CS has attracted wide attention and provides a new idea for solving the high-resolution ISAR imaging of high-speed maneuvering targets [13,14].

Among the CS algorithms, sparse Bayesian learning (SBL) [15–17] is widely used with high accuracy and strong robustness. SBL is a very important optimization algorithm based on Bayesian theory and statistics. In SBL, in order to improve signal sparsity, a hierarchical model is used to model the signal, that is, the prior information of the signal. Then, based on prior information and known observed data, the posterior distribution of the signal is solved using a Bayesian formula. The optimal mean value of the posterior distribution obtained by iteration is the reconstructed signal value is then finally obtained. However, the solution of the covariance matrix and the mean value of the posterior distribution involves a matrix inversion operation, and the calculation amount of direct inversion is proportional to the cube of the observed signal dimension. The large calculation amount hinders the development of the practical application of SBL.

In view of the shortcomings of SBL, many scholars have proposed some fast SBL algorithms. Therein, two low-complexity belief propagation (BP)-mean field (MF) SBL algorithms have been proposed [18]. In [19–21], an efficient SBL algorithm was proposed, which used a surrogate function to approximate the posterior density and to avoid matrix inversion. Moreover, approximate message passing (AMP) and generalized approximate message passing (GAMP) techniques were used to compute approximate marginal posteriors with a very low complexity to circumvent matrix inversion [22,23]. These algorithms all use some low-complexity methods instead of matrix inversion to reduce the computation; however, they all use approximation, resulting in results inferior to the original SBL. In order not to sacrifice the accuracy of SBL, a fast SBL without approximation is proposed in [24], which takes advantage of the special structure of the matrix to be inverse and uses Gohberg–Semencul (GS) decomposition [25,26] to obtain the decomposition formula of the inverse matrix. Then, based on the GS decomposition, the expression involving the inverse matrix can be calculated using fast Fourier, which greatly improves the computational efficiency. To extend this idea, a new fast free-approximation SBL algorithm is proposed in this paper, which uses a new decomposition method to obtain the inverse matrix. Both theoretical derivation and experimental simulation prove that the proposed algorithm has a lower computational complexity.

Motion compensation is the premise and basis of target ISAR imaging, and the compensation accuracy greatly affects the focus and sharpness of the image [27,28]. Motion compensation consists of envelope alignment and phase compensation. The traditional envelope alignment method is still suitable for maneuvering targets. When the radar resolution is not high, the rotation of the target in a short time can be regarded as uniform rotation, and a good image can be obtained through a traditional phase compensation technology, such as phase gradient autofocus (PGA) [29]. However, due to the maneuvering characteristics of the maneuvering target, the traditional phase compensation methods

are not very effective. Because an observation period is very short, in the radar signal processing cycle, the radar echo of a high-speed maneuvering target can be represented by a multi-component polynomial phase signal. When the target has a maneuvering motion with constant acceleration, the target echo signal can be expressed as the signal of linear frequency modulated (LFM) [30–34]. For more complex maneuvering targets, such as roll, pitch and yaw, the target echo signal is modeled as the cubic phase signal (CPS) [35–39]. Estimating and compensating for the parameters of the multi-component polynomial signal by signal processing has become the key to obtaining a high-quality and high-resolution ISAR image.

At present, studies on the parameter estimation of the LFM signal are relatively mature. Commonly used algorithms include Wigner–Ville distribution (WVD) [40], Wigner–Hough transform (WHT) [41], modified WVD (M-WVD) [33], Lv's distribution (LVD) [34], etc. However, there are few studies on the phase parameter estimation of the CPS signal, and many applications include a high-order phase compensation algorithm (PHMT) [42], high-order cubic phase compensation algorithm (PGCPF) [43], improved discrete Fourier algorithm (MDCFT) [35], etc. Although these algorithms improve the imaging quality of complex maneuvering targets, these algorithms are based on signal decomposition, which is sensitive to noise and requires a high amount of computation.

In ISAR imaging, the entropy of a two-dimensional image is a typical index for measuring the image sharpness and focusing effect. The lower the entropy, the better the image focus. Therefore, a new parameter estimation method based on minimum entropy [44,45] and a quasi-Newton algorithm [46] is proposed in this paper to transform the phase error compensation problem into a minimum entropy optimization problem. With the accumulation of energy, when the optimal solution is reached, the image entropy mainly comes from the strong scattering points, and the image becomes clearer. The randomness of noise means it has little effect on image entropy. Therefore, the minimum entropy algorithm can be used to estimate the phase error parameters more accurately. To solve the optimization problem, we use the quasi-Newton algorithm to find the optimal solution. In addition, the parameter estimation is combined with the proposed fast SBL algorithm for high-resolution imaging, that is, the high-order motion parameters of the target are accurately estimated in the iterative process of SBL.

The overall framework of this article is organized as follows. First, we give the notations of this article. The ISAR high-speed target signal model is briefly described in Section 2. In Section 3, we introduce in detail a new fast SBL algorithm and the minimum entropy algorithm to achieve high-resolution imaging and parameter estimation in imaging. Then, we validate the proposed methods using several numerical simulation results and discussions in Section 4. Finally, some conclusions are given in Section 5.

*Notations*

In this paper, matrices, vectors and scalars are denoted by uppercase boldface letters, lowercase boldface letters and italic letters. $(\cdot)^{-1}$, $(\cdot)^{H}$, $(\cdot)^{T}$ and $(\cdot)^{*}$ represent matrix inverse operation, conjugate transpose operation, transpose operation and conjugate operation, respectively. $diag(\cdot)$ denotes the main diagonal elements of a matrix. $\mathcal{CN}(\mathbf{x}|\mathbf{0}, \mathbf{A})$ denotes complex Gaussian distribution with mean $\mathbf{0}$ and covariance $\mathbf{A}$. $Gamma(\gamma|a, b)$ denotes that $\gamma$ follows the Gamma distribution with shape parameter a and scale parameter b. $(\cdot)^{(i)}$ represents the value obtained after the $i$th iteration. $[\cdot]_{M \times K}$ denotes that the dimension of a matrix is $M \times K$. $[\cdot]_{M}$ denotes that the dimension of a matrix is $M \times M$ or the dimension of a vector is $M$. $\mathbf{I}$ and $\mathbf{0}$ denote the proper size identity matrix and zero matrix/vector, respectively. For the vector $\mathbf{x} = [x_0, x_1, \cdots x_{n-1}]$, $x_i$ means the $(i+1)th$ element of $\mathbf{x}$. $\mathrm{Re}[\cdot]$ and $\mathrm{Im}[\cdot]$ denote taking the real part operator and taking imaginary part operator, respectively.

## 2. ISAR High-Speed Target Signal Model

Suppose that the radar transmits a LFM signal, after demodulation and range compression, the radar receiving signal of a moving target is

$$s(t_r, t_m) = \sum_{i=1}^{I} \delta_i \sin c \left[ B_r \left( t_r - \frac{2R_i(t_m)}{c} \right) \right] \exp \left( -j \frac{4\pi}{\lambda} R_i(t_m) \right) \tag{1}$$

where $t_r$ denotes the range fast-time variable. $t_m$ denotes the azimuth slow-time variable. $\delta_i$ denotes the complex reflectivity of the $i$th scattering point of moving target after pulse compression. I is the total number of scattered points of the target. $B_r$ is the bandwidth of the transmitted signal. $c$ is the speed of light. $\sin c(x) = \frac{\sin(\pi x)}{\pi x}$ represents the sinc function. $\lambda$ represents the wavelength of the signal. $R_i(t_m)$ represents the instantaneous distance between the radar and the $i$th scattering point on the target during the observation time. The movement of the target relative to the radar can be divided into two parts: translation and rotation; so, $R_i(t_m)$ has the form

$$R_i(t_m) = R_i^{trans}(t_m) + R_i^{rot}(t_m) \tag{2}$$

where $R_i^{rot}(t_m)$ represents the instantaneous distance caused by the rotation of the target. $R_i^{trans}(t_m)$ represents the instantaneous distance caused by the translation of the target, which is consistent for all scattered points on the target. $R_i^{trans}(t_m)$ can be expressed as $R_i^{trans}(t_m) = R_o + r(t_m)$. $R_0$ denotes the initial radar distance at the target center. $r(t_m)$ represents the instantaneous translational distance of the target.

According to the Weierstrass approximation theorem [47], the instantaneous distance of the target's translation in the accumulation time can be expressed as a polynomial function related to the slow time, i.e.,

$$r(t_m) = vt_m + a_1 t_m^2 + a_2 t_m^3 + \cdots \tag{3}$$

where $v$ denotes the target radial velocity. $a_1$ and $a_2$ denote the target acceleration and acceleration rate, respectively. Since the cubic term is already relatively close to the actual movement of the target, we do not consider the more than cubic term.

When the pulse accumulation time is short and the rotation angle of the target is small. The target can be regarded as rotating at a constant speed. The distance generated by the rotation of the target can be expressed as

$$\begin{aligned} R_i^{rot}(t_m) &= y_i \cos \theta(t_m) + x_i \sin \theta(t_m) \\ &\approx y_i + x_i w t_m \end{aligned} \tag{4}$$

where $\theta(t_m)$ denotes the instantaneous rotation angle of the target. With the target center as the reference point, the coordinate of the $i$th scattering point is $(x_i, y_i)$. $w$ denotes the target rotational angular speed.

Therefore, the instantaneous distance between the $i$th scattering point on the target and the radar can be expressed as

$$R_i(t_m) = R_o + vt_m + a_1 t_m^2 + a_2 t_m^3 + y_i + x_i w t_m \tag{5}$$

Substituting (5) into (1), the discrete form of echo signal after range migration compensation can be written as

$$s(n, m) = \widetilde{s}(n, m) \exp \left( -j \frac{4\pi}{\lambda} \left( v \frac{m}{PRF} + a_1 \left( \frac{m}{PRF} \right)^2 + a_2 \left( \frac{m}{PRF} \right)^3 \right) \right) \tag{6}$$

where $\widetilde{s}(n,m)$ is the ideal discrete echo form of the target and has the form

$$\widetilde{s}(n,m) = \sum_{i=1}^{I} \delta_i \sin c \left[ B_r \left( n \frac{T_{pulse}}{N} - \frac{2y_i}{c} \right) \right] \exp \left( -j \frac{4\pi}{\lambda} (R_0 + y_i) \right) \exp \left( -j \frac{4\pi}{\lambda PRF} x_i wm \right) \quad (7)$$

*n* and *m* denote range unit and pulse number, respectively. *N* and *M* denote the total number of range units and total number of pulses, respectively. *PRF* represents pulse repetition frequency (PRF) of the radar signal. From (6), we can see that the phase change generated by the scattering point movement on the maneuvering target can be approximately regarded as a polynomial, and the echo signal is a polynomial phase signal. However, these motion parameters in the polynomial are unknown, and so the parameter values should be estimated and compensated before imaging. The compensation accuracy will affect the imaging quality.

The one-dimensional range profile sequence shown in (6) can be written in matrix form after noise is added.

$$\mathbf{Y}_{\cdot n} = \mathbf{EFX}_{\cdot n} + \mathbf{\eta}_n \quad (8)$$

where $\mathbf{Y}_{\cdot n} \in \mathbf{C}^{M \times 1}$, $\mathbf{X}_{\cdot n} \in \mathbf{C}^{K \times 1}$ and $\mathbf{\eta}_n \in \mathbf{C}^{M \times 1}$ denote the one-dimensional range image, ISAR image and noise of the *n*th range unit, respectively. *K* represents the total number of image Doppler units. $\mathbf{E} \in \mathbf{C}^{M \times M}$ denotes the motion error of the target and is a diagonal matrix, in which the *m*th diagonal element denotes the motion error of the *m*th pulse, i.e., $\mathbf{E}_{m,m} = \exp \left( -j4\pi \left( v \frac{m}{PRF} + a_1 \left( \frac{m}{PRF} \right)^2 + a_2 \left( \frac{m}{PRF} \right)^3 \right) \right)$. $\mathbf{F} \in \mathbf{C}^{M \times K}$ denotes an overcomplete Fourier matrix with $M < K$ and has the form

$$\mathbf{F} = \begin{bmatrix} w_K^0 & w_K^0 & \cdots & w_K^0 \\ w_K^0 & w_K^1 & \cdots & w_K^{K-1} \\ \vdots & \vdots & \ddots & \vdots \\ w_K^0 & w_K^{M-1} & \cdots & w_K^{(M-1)(K-1)} \end{bmatrix}, w_K^m = \exp \left( -j \frac{2\pi}{K} k \right), k = 0, 1, 2, \cdots, K-1 \quad (9)$$

As shown in (8), the imaging process of an ideal echo is that $\mathbf{Y}$ and $\mathbf{F}$ are known to solve signal $\mathbf{X}$, and $\mathbf{X}$ is known to have a certain sparsity, that is, most elements in $\mathbf{X}$ are zero. But, the actual echo has a phase error, which will lead to the defocusing of the ISAR image. In order to achieve high-resolution imaging, it is necessary to estimate the parameters of phase errors and compensate them in imaging. Therefore, the imaging algorithm and parameter estimation are the key factors affecting imaging results.

Among many signal reconstruction algorithms, SBL has received more and more attention in recent years in the field of radar imaging because of its high accuracy in signal recovery. SBL makes full use of the prior information of parameters and sample information, and reconstructs signals through continuous iterative learning, so the reconstruction results are of high precision. In addition, SBL has strong a robustness and can considerably suppress the influence of noise. However, the high computational complexity of SBL hinders its development. In this paper, we propose a fast SBL algorithm, FSBL-LC for short, in which the time-consuming operations in SBL are replaced by some operations with a low computational complexity, and most of the operations can be solved using FFT, which greatly reduces the computational complexity.

Parameter estimation is generally divided into rough estimation and accurate estimation compensation. Rough estimation is used to estimate the initial value of parameters more accurately before imaging, and accurate estimation compensation is used to estimate the accurate value of parameters in imaging and make compensation. In this paper, a minimum entropy algorithm and fast SBL algorithm are combined for accurate parameter estimation and ISAR high-resolution imaging. These will be elaborated on in detail in the following section.

### 3. Fast SBL and High-Order Motion Parameter Estimation

In order to solve the defocusing problem of an ISAR image of a high-speed target, a novel high-resolution imaging algorithm with parameter-estimation-based SBL and a minimum entropy algorithm is proposed, which uses a new fast SBL algorithm to achieve high-resolution imaging, and uses the minimum Tsallis entropy algorithm to estimate the motion parameters of the target in the echo signal. In the following part, we will introduce the proposed fast SBL algorithm, the minimum Tsallis entropy algorithm, and how to use them to achieve high-resolution imaging of high-speed moving targets in more detail.

*3.1. The Proposed FSBL-LC Algorithm*

The mathematical model of sparse signal reconstruction is

$$\mathbf{y} = \mathbf{D}\mathbf{x} + \mathbf{\eta} \tag{10}$$

where $\mathbf{y} \in \mathbf{C}^{M \times 1}$, $\mathbf{D} \in \mathbf{C}^{M \times K}$, $\mathbf{x} \in \mathbf{C}^{K \times 1}$, and $\mathbf{\eta} \in \mathbf{C}^{M \times 1}$ are the measurement vector, the overcomplete dictionary matrix with $M < K$, the sparse signal vector to be recovered, and noise vector, respectively.

SBL adopts the hierarchical prior model. The first layer is the prior information of the signal. It is assumed that the signal and noise follow complex Gaussian distribution and their probability density functions (PDF) are, respectively,

$$p(\mathbf{x}) = \mathcal{CN}(\mathbf{x}|\mathbf{0}, \mathbf{A}) \tag{11}$$

$$p(\mathbf{\eta}) = \mathcal{CN}(\mathbf{\eta}|\mathbf{0}, \beta^{-1}\mathbf{I}) \tag{12}$$

where $\mathbf{A}$ is a diagonal matrix for which the diagonal elements are $\gamma_k^{-1}$ in sequence. $\gamma_k$ denotes the precision (inverse variance) of $x_k$. $\beta$ denotes the precision (inverse variance) of $\eta_k$. $x_k$ and $\eta_k$ are the $(k + 1)$th element of $\mathbf{x}$ and $\mathbf{\eta}$, respectively. $\gamma_k$ and $\beta$ are the hyperparameters. $\mathcal{CN}(\mathbf{x}|\mathbf{0}, \mathbf{A})$ denotes complex Gaussian distribution with mean $\mathbf{0}$ and covariance $\mathbf{A}$ and has the form $\mathcal{CN}(\mathbf{x}|\mathbf{0}, \mathbf{A}) = \pi^{-K} \prod_{k=0}^{K-1} \gamma_k \exp(-\gamma_k x_k^2)$.

The second layer is the prior information of the hyperparameters. Assume that they obey the Gamma distribution and their PDFs are

$$p(\gamma_k) = Gamma(\gamma_k|a, b) \tag{13}$$

$$p(\beta) = Gamma(\beta|c, d) \tag{14}$$

where $a$ and $b$ are the shape and scale parameters of $\gamma_k$, respectively. $c$ and $d$ are the shape and scale parameters of $\beta$, respectively. $Gamma(\gamma_k|a, b)$ denotes Gamma distribution and has the form $Gamma(\gamma_k|a, b) = \frac{b^a}{\Gamma(a)} \gamma_k^{a-1} \exp(-\gamma_k b)$, where $\Gamma(a)$ denotes the Gamma function.

Based on the above prior information, the posterior distribution of a signal can be obtained using a method similar to [15]:

$$p(\mathbf{x}|\mathbf{y}) = \mathcal{CN}(\mathbf{\mu}, \mathbf{\Sigma}) = \frac{1}{\pi^K|\mathbf{\Sigma}|} \exp\left(-(\mathbf{x} - \mathbf{\mu})^H \mathbf{\Sigma}^{-1}(\mathbf{x} - \mathbf{\mu})\right) \tag{15}$$

where the covariance and mean are

$$\mathbf{\Sigma} = \left(\beta \mathbf{D}^H \mathbf{D} + \mathbf{A}^{-1}\right)^{-1} \tag{16}$$

$$\mathbf{\mu} = \beta \mathbf{\Sigma} \mathbf{D}^H \mathbf{y} \tag{17}$$

SBL is used to obtain the reconstruction signal by seeking the optimal mean through continuous iteration. The updated covariance matrix and mean value in the iterative process are shown in (16) and (17). The iterative formulas for hyperparameters can be obtained by utilizing a similar procedure as in [15]:

$$\gamma_k^{(i+1)} = \frac{1 - \gamma_k^{(i)} \varepsilon_k^{(i)} + a}{\left(\mu_k^{(i)}\right)^2 + b} \tag{18}$$

$$\beta^{(i+1)} = \frac{N - \sum_{k=0}^{K-1} \alpha_k^{(i)} + c}{\|\mathbf{y} - \mathbf{D}\boldsymbol{\mu}^{(i)}\|_2^2 + d} \tag{19}$$

where $\varepsilon^{(i)} = diag(\Sigma^{(i)})$ and the $(\bullet)^{(i)}$ denotes the value after the *i*th iteration.

From the above principles and specific iterative steps of SBL, we know that the key step of single iteration is to solve $\varepsilon$ and $\boldsymbol{\mu}$, but their calculation involves matrix inversion and some matrix multiplication operations involving an inverse matrix, which increases the computational burden of SBL. However, we find that in ISAR imaging, the matrix to be inverted is a Toeplitz matrix, so the inverse matrix and matrix multiplication involving the inverse matrix can be quickly calculated using some product of Toeplitz matrices with cyclic matrices and FFT, respectively. The following is a detailed introduction.

The parameters in the ISAR imaging model shown in (8) are substituted into (16) and (17) yielding

$$\Sigma = \left(\beta(\mathbf{EF})^H \mathbf{EF} + \mathbf{A}^{-1}\right)^{-1} = \left(\beta \mathbf{F}^H \mathbf{F} + \mathbf{A}^{-1}\right)^{-1} = \mathbf{A} - \beta \mathbf{AF}^H \mathbf{R}^{-1} \mathbf{FA} \tag{20}$$

$$\boldsymbol{\mu}_{\cdot n} = \beta \Sigma \mathbf{F}^H \mathbf{E}^H \mathbf{Y}_{\cdot n} = \beta \mathbf{AF}^H \mathbf{R}^{-1} \mathbf{E}^H \mathbf{Y}_{\cdot n} \tag{21}$$

In (20) and (21), we use the Woodbury matrix identity to rewrite the covariance and mean, where $\mathbf{R} = \mathbf{I} + \beta \mathbf{FAF}^H$. We find the fact that $\mathbf{Q} = \mathbf{FAF}^H$ is a Hermitian–Toeplitz matrix that has the form

$$\mathbf{Q} = \begin{bmatrix} q_0 & q_1^* & \cdots & q_{M-1}^* \\ q_1 & q_0 & \cdots & q_{M-2}^* \\ \vdots & \vdots & \ddots & \vdots \\ q_{M-1} & q_{M-2} & \cdots & q_0 \end{bmatrix} \tag{22}$$

with

$$q_m = \sum_{k=0}^{K-1} \frac{1}{\gamma_k} \exp(-j2\pi mk/K) \tag{23}$$

It can be seen that elements in $\mathbf{Q}$ can be solved quickly using FFT. Therefore, $\mathbf{R}$ is also a Hermitian–Toeplitz matrix with the same structure as $\mathbf{Q}$ and its elements are also obtained through $K$-point FFT.

Since $\mathbf{R}$ is an $M \times M$ Hermitian-Toeplitz matrix, it can be rewritten as

$$\mathbf{R}_M = \begin{bmatrix} r_0 & \mathbf{r}_{M-1}^H \\ \mathbf{r}_{M-1} & \mathbf{R}_{M-1} \end{bmatrix} = \begin{bmatrix} \mathbf{R}_{M-1} & \widetilde{\mathbf{r}}_{M-1}^* \\ \widetilde{\mathbf{r}}_{M-1}^T & r_0 \end{bmatrix} \tag{24}$$

where $r_{M-1} = \begin{bmatrix} r_1 & r_2 & \cdots & r_{M-1} \end{bmatrix}^T$, $\widetilde{r}_{M-1} = \begin{bmatrix} r_{M-1} & r_{M-2} & \cdots & r_1 \end{bmatrix}^T$, $\mathbf{R}_{M-1}$ is a submatrix of $\mathbf{R}_M$ and has the Hermitian–Toeplitz structure.

According to the matrix inversion formula, $\mathbf{R}_M^{-1}$ has the form

$$\mathbf{R}_M^{-1} = \begin{bmatrix} 0 & \mathbf{0} \\ \mathbf{0} & \mathbf{R}_{M-1}^{-1} \end{bmatrix} + \frac{1}{t_{M-1}} \begin{bmatrix} 1 \\ \mathbf{p}_{M-1} \end{bmatrix} \begin{bmatrix} 1 & \mathbf{p}_{M-1}^H \end{bmatrix} \tag{25}$$

$$= \begin{bmatrix} \mathbf{R}_{M-1}^{-1} 0 & \mathbf{0} \\ \mathbf{0} & 0 \end{bmatrix} + \frac{1}{t_{M-1}} \begin{bmatrix} \widetilde{\mathbf{p}}_{M-1}^* \\ 1 \end{bmatrix} \begin{bmatrix} \widetilde{\mathbf{p}}_{M-1}^T & 1 \end{bmatrix} \tag{26}$$

where

$$t_{M-1} = r_0 - \mathbf{r}_{M-1}^H \mathbf{R}_{N-1}^{-1} \mathbf{r}_{N-1} \tag{27}$$

$$\mathbf{p}_{M-1} = -\mathbf{R}_{N-1}^{-1} \mathbf{r}_{N-1} \tag{28}$$

We define an $M \times M$ lower-triangular matrix in which the 1st main diagonal elements are one and all other elements are zero, as well as an $M \times M$ cyclic matrix, as

$$\mathbf{L}_M = \begin{bmatrix} 0 & 0 & \cdots & 0 \\ 1 & 0 & \cdots & 0 \\ \vdots & \ddots & \ddots & \vdots \\ 0 & \cdots & 1 & 0 \end{bmatrix}_M \tag{29}$$

$$\mathbf{C}_M = \begin{bmatrix} 0 & 0 & \cdots & 1 \\ 1 & 0 & \cdots & 0 \\ \vdots & \ddots & \ddots & \vdots \\ 0 & \cdots & 1 & 0 \end{bmatrix}_M \tag{30}$$

We can obtain the fact

$$\begin{bmatrix} 0 & \mathbf{0} \\ \mathbf{0} & \mathbf{R}_{M-1}^{-1} \end{bmatrix} = \mathbf{L}_M \begin{bmatrix} \mathbf{R}_{M-1}^{-1} & \mathbf{0} \\ \mathbf{0} & 0 \end{bmatrix} \mathbf{C}_M^T \tag{31}$$

Based on the fact in (31), the displacement representation of $\mathbf{R}_M^{-1}$ can be written as

$$\begin{aligned} \nabla \mathbf{R}_M^{-1} &= \mathbf{R}_M^{-1} - \mathbf{L}_M \mathbf{R}_M^{-1} \mathbf{C}_M^T \\ &= \begin{bmatrix} 0 & \mathbf{0} \\ \mathbf{0} & \mathbf{R}_{M-1}^{-1} \end{bmatrix} + \frac{1}{t_{M-1}} \begin{bmatrix} 1 \\ \mathbf{p}_{M-1} \end{bmatrix} \begin{bmatrix} 1 & \mathbf{p}_{M-1}^H \end{bmatrix} \\ &\quad - \mathbf{L}_M \left\{ \begin{bmatrix} \mathbf{R}_{M-1}^{-1} 0 & \mathbf{0} \\ \mathbf{0} & 0 \end{bmatrix} + \frac{1}{t_{M-1}} \begin{bmatrix} \widetilde{\mathbf{p}}_{M-1}^* \\ 1 \end{bmatrix} \begin{bmatrix} \widetilde{\mathbf{p}}_{M-1}^T & 1 \end{bmatrix} \right\} \mathbf{C}_M^T \\ &= \frac{1}{t_{M-1}} \begin{bmatrix} 1 \\ \mathbf{p}_{M-1} \end{bmatrix} \begin{bmatrix} 1 & \mathbf{p}_{M-1}^H \end{bmatrix} - \frac{1}{t_{M-1}} \begin{bmatrix} 0 \\ \widetilde{\mathbf{p}}_{M-1}^* \end{bmatrix} \begin{bmatrix} 1 & \widetilde{\mathbf{p}}_{M-1}^T \end{bmatrix} \end{aligned} \tag{32}$$

Given $\nabla \mathbf{R}_M^{-1}$, $\mathbf{R}_M^{-1}$ has the form

$$\begin{aligned} \mathbf{R}_M^{-1} &= \sum_{m=0}^{M-1} (\mathbf{L}_M)^m \nabla \mathbf{R}_M^{-1} \left( \mathbf{C}_M^T \right)^m \\ &= \sum_{m=0}^{M-1} (\mathbf{L}_M)^m \left( \mathbf{w}_M \mathbf{w}_M^H - \mathbf{v}_M \bar{\mathbf{v}}_M^H \right) \left( \mathbf{C}_M^T \right)^m \\ &= \mathcal{L}_M(\mathbf{w}_M, \mathbf{L}_M) \mathcal{C}_M^H(\mathbf{w}_M, \mathbf{C}_M) - \mathcal{L}_M(\mathbf{v}_M, \mathbf{L}_M) \mathcal{C}_M^H(\hat{\mathbf{v}}_M, \mathbf{C}_M) \end{aligned} \tag{33}$$

where $\mathbf{w}_M = \frac{1}{\sqrt{t_{M-1}}} \begin{bmatrix} 1 \\ \mathbf{p}_{M-1} \end{bmatrix}$, $\mathbf{v}_M = \frac{1}{\sqrt{t_{M-1}}} \begin{bmatrix} 0 \\ \widetilde{\mathbf{p}}_{M-1}^* \end{bmatrix}$, $\hat{\mathbf{v}}_M = \frac{1}{\sqrt{t_{M-1}}} \begin{bmatrix} 1 \\ \widetilde{\mathbf{p}}_{M-1}^* \end{bmatrix}$, $\mathcal{L}_M(\mathbf{w}_M, \mathbf{L}_M)$, and $\mathcal{C}_M(\mathbf{w}_M, \mathbf{C}_M)$ denote the lower triangular Toeplitz matrix and the cyclic matrix composed of the elements of $\mathbf{w}_M$ and have the form

$$\mathcal{L}_M(\mathbf{w}_M, \mathbf{L}_M) \triangleq \begin{bmatrix} \mathbf{w}_M & \mathbf{L}_M \mathbf{w}_M & \cdots & (\mathbf{L}_M)^{M-1} \mathbf{w}_M \end{bmatrix} \tag{34}$$

$$\mathcal{C}_M(\mathbf{w}_M, \mathbf{C}_M) \triangleq \begin{bmatrix} \mathbf{w}_M & \mathbf{C}_M \mathbf{w}_M & \cdots & (\mathbf{C}_M)^{M-1} \mathbf{w}_M \end{bmatrix} \tag{35}$$

From (33), the calculation of $\mathbf{R}_M^{-1}$ is converted to the product of a Toeplitz matrix with a cyclic matrix. We call this decomposition the L-C decomposition in the paper, and the resulting formula is called the L-C decomposition formula. $\mathbf{w}_M$ and $\mathbf{v}_M$ are called L-C factorization factors of $\mathbf{R}_M^{-1}$, which can be solved by applying the Levinson–Durbin (L–D)-type algorithm [24].

Based on the L-C decomposition formula of $\mathbf{R}_M^{-1}$ in (33), the key steps in SBL can be quickly calculated using FFT, which greatly improves computing efficiency.

From (20), we observe that $\mathbf{A}$ is a diagonal matrix, so $\boldsymbol{\varepsilon}$ can be obtained using the dot product of $\beta$, the diagonal elements of $\mathbf{A}$, and the diagonal elements of $\mathbf{G} = \mathbf{F}^H \mathbf{R}_M^{-1} \mathbf{F}$. Let $\mathbf{g}_K = diag(\mathbf{G}) = \begin{bmatrix} g_0 & g_1 & \cdots & g_{K-1} \end{bmatrix}^T$, namely $\mathbf{g}_K$ is a vector made up of diagonal elements of $\mathbf{G}$. By substituting (25) into $\mathbf{G}$, we can obtain

$$g_k = \mathbf{F}_M^H(\omega_k) \mathbf{R}_M^{-1} \mathbf{F}_M(\omega_k) = \sum_{m=-M+1}^{M-1} c_m \exp(-j2\pi mk/K) \tag{36}$$

where $\mathbf{F}_M(\omega_k)$ denotes the (k + 1)-th column of $\mathbf{F}_M$ and its form is shown in (9). $c_m$ is the sum of all the elements on the m-th diagonal in $\mathbf{R}_M^{-1}$. It can be seen from (36) that the $\mathbf{g}_K$ can be quickly calculated using FFT, and $c_m$ can also be obtained using FFT. Let $\hat{\mathbf{c}} = \begin{bmatrix} c_{-M+1} & \cdots & c_{-1} & c_0 \end{bmatrix}^T$, given the L-C factorization formula of $\mathbf{R}_M^{-1}$, we can obtain

$$\hat{\mathbf{c}} = \mathcal{L}_M(\overline{\mathbf{w}}_M, \mathbf{L}_M) \mathbf{w}_M^* - \mathcal{L}_M(\overline{\mathbf{v}}_M, \mathbf{L}_M) \mathbf{v}_M^* \tag{37}$$

where $\overline{\mathbf{w}}_M = \begin{bmatrix} w_{m-1} & 2w_{m-2} & \cdots & (M-1)w_1 & Mw_0 \end{bmatrix}^T$, $w_m$ is the $(m+1)$th element of $\mathbf{w}_M$. The annotation for $\overline{\mathbf{v}}_M$ is similar to that for $\overline{\mathbf{w}}_M$. Obviously, based on the L-C factorization factor of $\mathbf{R}_M^{-1}$, $\hat{\mathbf{c}}$ can be obtained using the products of the Toeplitz matrix and vector, avoiding the calculation of $\mathbf{R}_M^{-1}$. Fortunately, the product of the Toeplitz matrix and vector can be converted to a circulant matrix–vector product, and then it can be solved via circular convolution; so, $\hat{\mathbf{c}}$ can be solved using $(2M-1)$-point FFT/IFFT.

From (21), $\boldsymbol{\mu}_{\cdot n}$ can be solved in three steps: $\boldsymbol{\phi}_M = \mathbf{R}^{-1} \mathbf{E}^H \mathbf{Y}_{\cdot n}$, $\boldsymbol{\varphi}_K = \mathbf{F}^H \boldsymbol{\phi}_M$ and $\boldsymbol{\mu} = \beta \mathbf{A} \boldsymbol{\varphi}_K$. For $\boldsymbol{\phi}_M$, given the L-C decomposition formula of $\mathbf{R}^{-1}$ in (33), $\boldsymbol{\phi}_M$ can be calculated using two cyclic matrix–vector products and two Toeplitz matrix–vector products. As mentioned above, the product of a Toeplitz matrix with a vector can be quickly computed using FFT/IFFT, while the product of a cyclic matrix and a vector can be converted into cyclic convolution and can be quickly computed using FFT/IFFT. Therefore, $\boldsymbol{\phi}_M$ can be obtained quickly via FFT/IFFT. Given $\boldsymbol{\phi}_M$, we have $\boldsymbol{\varphi}_K$, which can be obtained using K-point IFFT. Finally, since $\mathbf{A}$ and $\mathbf{E}$ are diagonal matrices, $\boldsymbol{\mu}_{\cdot n}$ can be computed using the dot product of $\beta$, the diagonal elements of $\mathbf{A}$ and $\boldsymbol{\varphi}_K$.

The specific steps of the proposed FSBL-LC are as follows:

Input: $\mathbf{Y}_{\cdot n}$, $\mathbf{E}^0$.

Step 1: Initialization. Set the initial value of the hyperparameter $\gamma_k^{(0)} = 1$ and $\beta^{(0)} = 1$. $a = b = c = d = 10^{-6}$.

Step 2: Update $\gamma_k^{(i+1)}$ and $\beta^{(i+1)}$ according to (18) and (19). Note that Step 2 is not required for the first iteration with the initial value.

Step 3: Compute the first column of $\mathbf{R}_M$ according to (22) by using FFT with $\mathcal{O}(K \log_2 K)$, and compute the L-C factorization factors of $\mathbf{R}_M^{-1}$ by using the L–D algorithm with $\mathcal{O}(M^2)$.

Step 4: Update $\boldsymbol{\varepsilon}^{(i+1)}$ and $\boldsymbol{\mu}_{\cdot n}^{(i+1)}$ by using FFT with $\mathcal{O}(M \log_2 M + K \log_2 K)$.

Step 5: Let $i = i + 1$, return to Step 2, and continue the loop until the loop stops.

## 3.2. High-Order Motion Parameter Estimation Based on Minimum Entropy

In this paper, the parameter values of the high-order motion parameter of the target are accurately estimated in the iterative process of ISAR imaging. The accuracy of parameter estimation has a great influence on imaging results. Image entropy is a good measure of the focusing effect of the resulting image. So, the minimum entropy algorithm is often

used to accurately estimate the motion parameter of the target. Among the many entropy functions, Tsallis entropy is considered to be an application for explaining non-extensive system problems, and so it tends to have better results than other entropies.

The Tsallis entropy of the ISAR image obtained using the SBL algorithm is

$$T_{\mu_n} = \frac{1}{p-1} - \sum_{n=0}^{N-1} \sum_{k=0}^{K-1} \frac{1}{p-1} \left( \frac{|\mu_{k,n}|^2}{P_n} \right)^p \tag{38}$$

where $P_n$ is the total energy of the image of the $n$th range unit.

The target phase parameter estimation based on minimum Tsallis entropy can be expressed as

$$\{\hat{v}, \hat{a}_1, \hat{a}_2\} = \underset{v, a_1, a_2}{\mathrm{argmin}} \left( T_{\mu_n} \right) \tag{39}$$

where $\hat{v}$, $\hat{a}_1$ and $\hat{a}_2$ are the estimation values of the motion parameters of the target.

It can be seen from (39) that parameter estimation is transformed into an optimization problem. Among many algorithms for solving optimization problems, the quasi-Newton algorithm not only uses the gradient of objective function but also the second derivative property of objective function; so, it has a higher accuracy and faster convergence rate. BFGS correction is the most popular and effective quasi-Newtonian correction, and so it is widely used. In the BFGS algorithm, the gradient of the cost function corresponding to each parameter needs to be solved. We define a parameter vector $\theta$ containing the parameters to be estimated, namely $\theta = \begin{bmatrix} v & a_1 & a_2 \end{bmatrix}^T$. From (29), the gradient of image entropy $T_\mu$ with respect to $\theta$ is

$$\frac{\partial T_{\mu_n}}{\partial \theta} = \frac{\partial T_{\mu_n}}{\partial |\mu_{k,n}|^2} \frac{\partial |\mu_{k,n}|^2}{\partial \theta} = -\sum_{k=0}^{K-1} \sum_{n=0}^{N-1} \frac{p}{p-1} \frac{|\mu_{k,n}|^{2(p-1)}}{P^p} \frac{\partial |\mu_{k,n}|^2}{\partial \theta} \tag{40}$$

According to (21) and the high-order motion error **E**, we can obtain

$$\mu_{k,n} = \beta \gamma_k^{-1} \sum_{m=0}^{M-1} \left( \mathbf{F}^H \mathbf{R}_n^{-1} \right)_{k,m} \mathbf{Y}_{m,n} \exp \left( j \left( \frac{4\pi v}{\lambda PRF} m + \frac{4\pi a_1}{\lambda PRF2} m^2 + \frac{4\pi a_2}{\lambda PRF3} m^3 \right) \right) \tag{41}$$

Then, the gradient of $|\mu_{k,n}|^2$ with respect to $\theta$ is

$$\frac{\partial |\mu_{k,n}|^2}{\partial \theta} = 2\mathrm{Re} \left( \mu_{k,n}^* \frac{\partial \mu_{k,n}}{\partial \theta} \right) \tag{42}$$

Substituting (42) into (40) yields

$$\frac{\partial \mu_{k,n}}{\partial v} = j\beta \gamma_k^{-1} \sum_{m=0}^{M-1} \left( F^H R_n^{-1} \right)_{k,m} \mathbf{Y}_{m,n} \exp \left( j \left( \frac{4\pi v}{\lambda PRF} m + \frac{4\pi a_1}{\lambda PRF2} m^2 + \frac{4\pi a_2}{\lambda PRF3} m^3 \right) \right) \frac{4\pi m}{\lambda PRF} \tag{43}$$

$$\frac{\partial \mu_{k,n}}{\partial a_1} = j\beta \gamma_k^{-1} \sum_{m=0}^{M-1} \left( \mathbf{F}^H \mathbf{R}_n^{-1} \right)_{k,m} \mathbf{Y}_{m,n} \exp \left( j \left( \frac{4\pi v}{\lambda PRF} m + \frac{4\pi a_1}{\lambda PRF2} m^2 + \frac{4\pi a_2}{\lambda PRF3} m^3 \right) \right) \frac{4\pi m^2}{\lambda PRF2} \tag{44}$$

$$\frac{\partial \mu_{k,n}}{\partial a_2} = j\beta \gamma_k^{-1} \sum_{m=0}^{M-1} \left( \mathbf{F}^H \mathbf{R}_n^{-1} \right)_{k,m} \mathbf{Y}_{m,n} \exp \left( j \left( \frac{4\pi v}{\lambda PRF} m + \frac{4\pi a_1}{\lambda PRF2} m^2 + \frac{4\pi a_2}{\lambda PRF3} m^3 \right) \right) \frac{4\pi m^3}{\lambda PRF3} \tag{45}$$

So, the gradients of image entropy $T_\mu$ with respect to $v$, $a_1$, and $a_2$ are

$$\frac{\partial T_{\mu n}}{\partial v} = -\sum_{k=0}^{K-1}\sum_{n=0}^{N-1} \frac{2p}{p-1} \frac{|\mu_{k,n}|^{2(p-1)}}{P^p}$$
$$\cdot \mathrm{Im}\left( \mu_{k,n}^* \beta \gamma_k^{-1} \sum_{m=0}^{M-1} \left(\mathbf{F}^H \mathbf{R}_n^{-1}\right)_{k,m} \mathbf{Y}_{m,n} \exp\left( j\left( \frac{4\pi v}{\lambda PRF} m + \frac{4\pi a_1}{\lambda PRF^2} m^2 + \frac{4\pi a_2}{\lambda PRF^3} m^3 \right) \right) \frac{4\pi m}{\lambda PRF} \right) \tag{46}$$

$$\frac{\partial T_{\mu n}}{\partial a_1} = -\sum_{k=0}^{K-1}\sum_{n=0}^{N-1} \frac{2p}{p-1} \frac{|\mu_{k,n}|^{2(p-1)}}{P^p}$$
$$\cdot \mathrm{Im}\left( \mu_{k,n}^* \beta \gamma_k^{-1} \sum_{m=0}^{M-1} \left(\mathbf{F}^H \mathbf{R}_n^{-1}\right)_{k,m} \mathbf{Y}_{m,n} \exp\left( j\left( \frac{4\pi v}{\lambda PRF} m + \frac{4\pi a_1}{\lambda PRF^2} m^2 + \frac{4\pi a_2}{\lambda PRF^3} m^3 \right) \right) \frac{4\pi m^2}{\lambda PRF^2} \right) \tag{47}$$

$$\frac{\partial T_{\mu n}}{\partial a_2} = -\sum_{k=0}^{K-1}\sum_{n=0}^{N-1} \frac{2p}{p-1} \frac{|\mu_{k,n}|^{2(p-1)}}{P^p}$$
$$\cdot \mathrm{Im}\left( \mu_{k,n}^* \beta \gamma_k^{-1} \sum_{m=0}^{M-1} \left(\mathbf{F}^H \mathbf{R}_n^{-1}\right)_{k,m} \mathbf{Y}_{m,n} \exp\left( j\left( \frac{4\pi v}{\lambda PRF} m + \frac{4\pi a_1}{\lambda PRF^2} m^2 + \frac{4\pi a_2}{\lambda PRF^3} m^3 \right) \right) \frac{4\pi m^3}{\lambda PRF^3} \right) \tag{48}$$

Based on the gradients obtained above, the updated formulas for parameters estimation are

$$\hat{\theta}^{(j+1)} = \hat{\theta}^{(j)} - \lambda^{(j)} \left(\mathbf{B}^{(j)}\right)^{-1} \nabla T_\mu(\hat{\theta}^{(j)}) \tag{49}$$

where $\hat{\theta}$ and $\nabla T_\mu(\hat{\theta}^{(j)})$ denote the estimated value of $\theta$ and the gradient of the cost function with respect to $\hat{\theta}$. The superscript $(j)$ denotes the number of cycles. $\lambda^{(j)}$ is the search step vector corresponding to $\hat{\theta}^{(j)}$, which can be obtained via the Armijo criterion. $\mathbf{B}^{(j)}$ is the approximate matrix corresponding to $\hat{\theta}^{(j)}$, which is used to replace the Hessian matrix of the cost function.

The specific steps of accurately estimating the optimal parameters using the BFGS algorithm are as follows:

Step 1: Initialization. Set the number of cycles $j = 0$. Set the initial parameter value of the parameter to be estimated to the value obtained using the rough estimation, namely $\hat{\theta}^{(0)} = \hat{\theta}_{rough}$ ($\hat{\theta}_{rough}$ denotes the value obtained from the rough estimation). Set the approximate matrix $\mathbf{B}^{(0)} = \mathbf{I}_{3\times 3}$. And calculate the gradient $\nabla T_\mu(\hat{\theta}^{(0)})$ based on $\mathbf{B}^{(0)}$ and $\hat{\theta}^{(0)}$.

Step 2: The search step vector $\lambda^{(j)}$ is determined according to the Armijo criterion.

Step 3: Update $\hat{\theta}^{(j+1)}$ according to (49) and calculate $\mu_{\cdot n}$ by using the method described in FSBL-LC.

Step 4: Update $\nabla T_\mu(\hat{\theta}^{(j)})$ according to (47) and (48).

Step 5: Compute $\mathbf{B}^{(j+1)}$ according to the following formula.

$$\mathbf{B}^{(j+1)} = \begin{cases} \mathbf{B}^{(j)}, & \left(\mathbf{t}^{(j)}\right)^T \mathbf{s}^{(j)} \le 0 \\ \mathbf{B}^{(j)} - \dfrac{\mathbf{B}^{(j)}\mathbf{s}^{(j)}\left(\mathbf{s}^{(j)}\right)^T \mathbf{B}^{(j)}}{\left(\mathbf{s}^{(j)}\right)^T \mathbf{B}\mathbf{s}^{(j)}} + \dfrac{\mathbf{t}^{(j)}\left(\mathbf{t}^{(j)}\right)^T}{\left(\mathbf{t}^{(j)}\right)^T \mathbf{s}^{(j)}}, & \left(\mathbf{t}^{(j)}\right)^T \mathbf{s}^{(j)} > 0 \end{cases} \tag{50}$$

where $\mathbf{s}^{(j)} = \theta^{(j+1)} - \theta^{(j)}$, $\mathbf{t}^{(j)} = \nabla T_\mu(\hat{\theta}^{(j+1)}) - \nabla T_\mu(\hat{\theta}^{(j)})$.

Step 6: Let $j = j + 1$, return to Step 2, and continue the loop until the loop stops.

The flow chart of high-resolution imaging, with embedded parameter estimation based on FSBL-LC and the quasi-Newton algorithm, is shown in Figure 1.

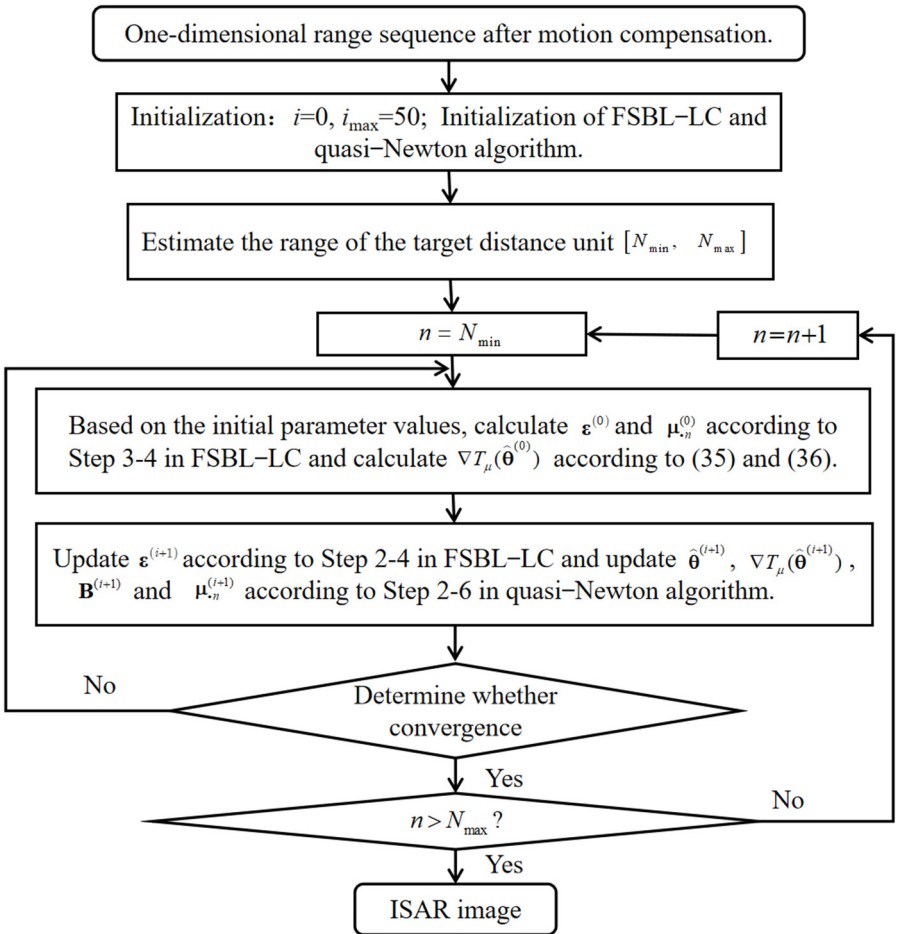

**Figure 1.** The imaging process based on the proposed method.

In the above imaging process, we find that the calculation of $\boldsymbol{\mu}_{\cdot n}$ and $\nabla T_\mu(\hat{\boldsymbol{\theta}}^{(j)})$ in the parameter estimation section contains the multiplication of $\mathbf{R}_n^{-1}$ by the vector and Fourier dictionary by the vector, which can be quickly calculated using FFT/IFFT based on the L-C decomposition formula of $\mathbf{R}_n^{-1}$. In addition, in parameter estimation, rough estimation is required first, because, for optimization algorithms, the initial value close to the true value is very beneficial for finding the optimal parameters; so, it is necessary to carry out a rough estimation of parameters before the accurate estimation of parameters, which does not only improve the convergence efficiency, but also ensures the global optimal solution.

## 4. Simulation Processing Result

This section consists of two parts. Since our article focuses on high-resolution imaging methods, we first verified the performance of the proposed FSBL-LC algorithm using simulation experiments. Then, we presented the simulation and measured data experiments of maneuvering target parameter estimation and high-resolution imaging based on the minimum entropy algorithm and FSBL-LC.

### 4.1. Performance of FSBL-LC Algorithm

In this subsection, we only verify the FSBL-LC algorithm separately. We show the one-dimensional signal reconstruction and the algorithm performance comparison graph. The algorithms compared with the proposed FSBL-LC include the original SBL algorithm (DI-SBL for short) and FSBL-GS (FSBL-GS is an approximation-free fast SBL algorithm proposed in the literature [24]), which is abbreviated as FSBL-GS here. The difference between FSBL-GS and the algorithm introduced in this paper lies in the different decomposition forms of the inverse matrix. Since FSBL-GS has been compared with some reconstruction algorithms

and the typical proposed fast SBL algorithm using approximations in the literature [24], this paper is just a simple comparison with FSBL-GS. The hyperparameters are set to $a = b = c = d = 10^{-6}$. The initial values of $\gamma$ and $\beta$ are 1. The condition for SBL iteration to stop is that the following convergence conditions are met:

$$\frac{\|\boldsymbol{\mu}_{.n}^{(j)} - \boldsymbol{\mu}_{.n}^{(j-1)}\|}{\|\boldsymbol{\mu}_{.n}^{(j)}\|} \leq \delta \tag{51}$$

where $\delta$ denotes the convergence threshold and is set to $10^{-4}$.

In addition, the normalization error is defined as the qualitative comparative reconstruction accuracy.

$$\text{nRMSE} = \frac{\|\hat{\mathbf{x}} - \mathbf{x}\|_2}{\|\mathbf{x}\|_2} \tag{52}$$

where $\hat{\mathbf{x}}$ denotes the reconstructed signal and $\mathbf{x}$ denotes the real signal.

In Figure 2, we show the reconstruction results of a one-dimensional signal. In the experiment, the simulated data are derived from a one-dimensional complex signal with six frequency components. The SNR is set at 10 dB, the length of the observed data is 512, and the super-resolution factor (SRF = K/M) is 4. Figure 2a shows the reconstruction result of FFT, and Figure 2b shows the reconstruction results of DI-SBL, FSBL-GS, and FSBL-LC. The red dots represent the real signal, and the blue line represents the reconstructed signal. It can be seen from Figure 2 that the sidelobe of the FFT result is higher, and the results of FSBL-GS, FSBL-LC, and DI-SBL are the same; because FSBL-GS and FSBL-LC do not adopt approximation, the calculation speed is improved without sacrificing the reconstruction accuracy.

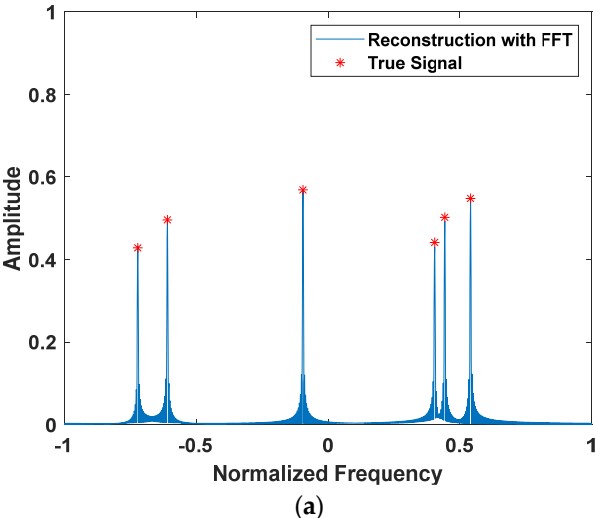 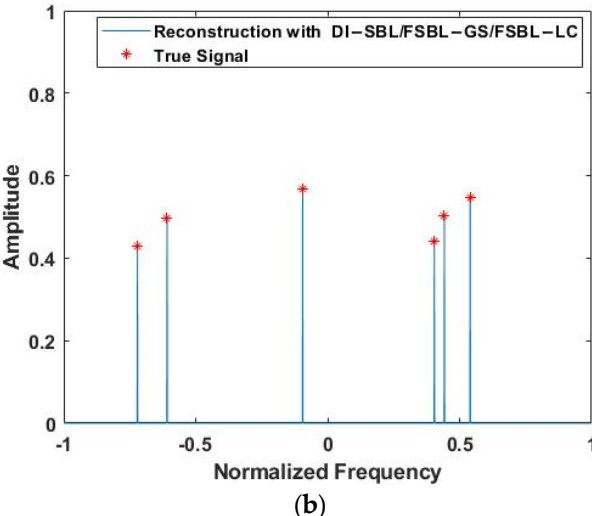

**Figure 2.** Signal reconstruction results provided by (**a**) FFT and (**b**) DI-SBL/FSBL-GS/FSBL-LC.

Table 1 shows the mean computation time and nRMSE for 500 Monte Carlo experiments. The experiments used random signals. As can be seen from Table 1, the calculation time of FSBL-LC proposed in this paper is the shortest, and the calculation efficiency is 46 times higher than that of DI-SBL and 2 times higher than that of FSBL-GS. DI-SBL requires the computation of $\mathbf{R}^{-1}$ and some multiplication operations involving the inverse matrix; so, the computational complexity is very high. Both FSBL-GS and FSBL-LC algorithms solve the decomposition formula of $\mathbf{R}^{-1}$ so as to replace the direct inverse of the matrix. They also use FFT to quickly solve the multiplication operation involving the inverse matrix. So, the computational complexity is reduced and the computational efficiency is improved. Through observation, we can see that the difference between FSBL-GS and FSBL-LC is that the decomposition formula of $\mathbf{R}^{-1}$ is different. When solving $\boldsymbol{\phi}_M$ in the

first step of solving $\boldsymbol{\mu}_{\cdot,n}$, the FSBL-GS algorithm uses the product of the Toplitz matrix and vector to convert them into a high-dimensional cyclic matrix and vector product, and then uses $(2M - 1)$ point FFT and IFFT to calculate this quickly. FSBL-LC calculates $\boldsymbol{\phi}_M$ using the product of cyclic matrix and vector, which can be quickly calculated using the $M$ point FFT and IFFT. It can be seen that FSBL-LC has a lower computational complexity.

**Table 1.** The mean computation time and nRMSE.

|  | **DI-SBL** | **FSBL-GS** | **FSBL-LC** |
| --- | --- | --- | --- |
| Time(s) | 8.4567 | 0.3640 | 0.1815 |
| nRMSE | 0.0226 | 0.0226 | 0.0226 |

Next, in Figures 3 and 4, we give the algorithm performance comparison diagram with various SNR, lengths of observation window M, and SRF. In the experiment showing the effect of SNR, the observation window of the simulation data is 256 and SRF is 4. In the experiment showing the effect of M and SRF, SNR is set to 10 dB. Figure 3a shows the effect of the calculation time on SNR, and Figure 3b is the reconstruction error diagram at different SNR. It can be seen from the figure that the reconstruction results of DI-SBL, FSBL-GS, and FSBL-LC are the same, and so the error curves coincide. The larger the SNR, the smaller the reconstruction time and the smaller the reconstruction error. Compared with the original SBL, the calculation efficiency of FSBL-GS and FSBL-LC is significantly improved, and the calculation time of FSBL-LC is shorter than that of FSBL-GS. When SNR is 20 dB, the error value is small, almost close to zero. Figure 4 is the calculation time diagram with different M and SRF. The red line and dots represent the calculation time of the DI-SBL. The blue line and dots represent the calculation time of the FSBL-GS. The pink lines and dots represent the calculation time of the FSBL-LC. As can be seen from the figure, as the length of the observation window becomes longer, the reconstruction takes longer, and as the value of the super-resolution factor increases, so does the time.

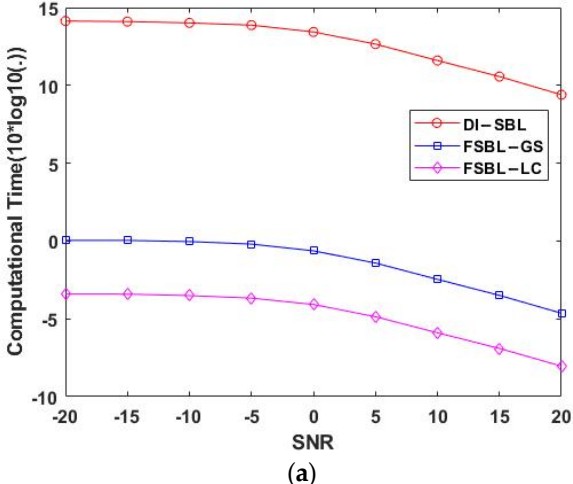
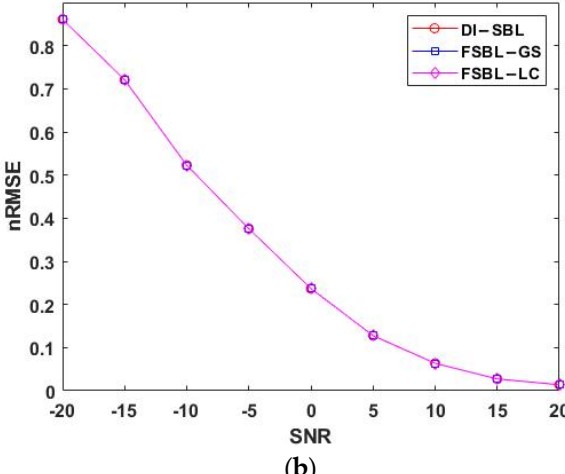

(**a**)          (**b**)

**Figure 3.** Average computational times and nRMSE of algorithms with SNR. (**a**) Average computational times of algorithms. (**b**) nRMSE of algorithms.

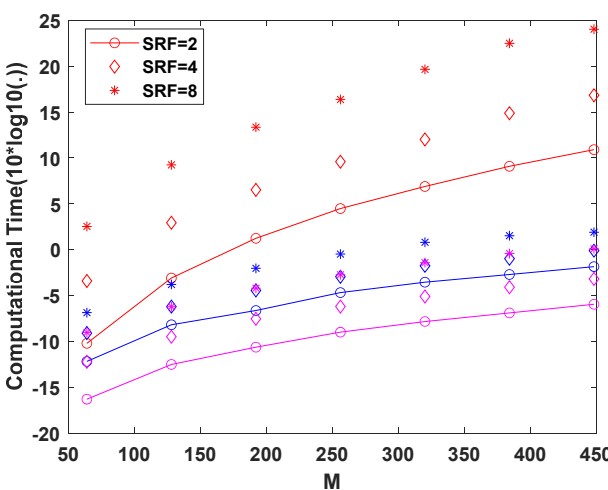

**Figure 4.** Average computational times of algorithms with M and SRF. The red line and dots represent the calculation time of the DI-SBL. The blue line and dots represent the calculation time of the FSBL-GS. The pink lines and dots represent the calculation time of the FSBL-LC.

### 4.2. Maneuvering Target Parameter Estimation and High-Resolution Imaging

After comparing the performance of the proposed FSBL-LC, we used the imaging process shown in Figure 1 to achieve error parameter estimation and high-resolution imaging of high-speed maneuvering targets in this subsection. The radar system and target parameters in all experiments in this subsection are shown in Table 2. Motion compensation was used before imaging. In the experiments, the minimum entropy of the image is taken as the index of image quality, and the parameter rough estimation is realized by using an exhaustive linear search.

**Table 2.** Radar system and target parameters.

| Parameter | Value | Parameter | Value |
|---|---|---|---|
| Carrier frequency | 20 GHz | Bandwidth | 1 GHz |
| Pulse width | 500 us | Pulse repetition frequency | 200 Hz |
| Initial distance between target center and radar | 600 KM | Rotational angular speed | 0.015 rad/s |
| Range dimension sampling number | 512 | Azimuth dimension sampling number | 1024 |

Firstly, we use simulation data from a simple satellite model to verify the validity of the proposed parameter estimation and high-resolution imaging method. We model the satellite as a high-speed maneuvering target with a non-stationary flight motion of uniform rotation. Translational motion has an acceleration and an acceleration rate, that is, the satellite's translational motion is expressed as a third-order polynomial. Assume that the satellite's translational velocity is 10 KM/s, its translational acceleration is 120 M/s$^2$, and its translational acceleration rate is 40 M/s$^3$. We added noise to a one-dimensional-range image and set the SNR to 10 dB.

The one-dimensional-range image and imaging results of algorithms are shown in Figure 5. Figure 5a shows the ideal satellite imaging result, namely the RD image result when the target is flying smoothly. Figure 5b is the one-dimensional-range image of the target in non-stationary flight. It can be seen from the figure that the maneuvering characteristics of the target make the range image appear migration through range cell. The one-dimensional-range image after envelope alignment, using the traditional minimum entropy algorithm, is shown in Figure 5c. Figure 5d shows the imaging results obtained

using FFT on the one-dimensional-range image after envelope alignment in the azimuth dimension. It can be seen from the figure that the RD has been basically invalid due to the phase error caused by the target motion characteristics. Figure 5e is the imaging result obtained using FFT in the azimuth dimension after phase error correction based on rough estimates. Since the rough estimate is a value close to the true value and RD has a low resolution, the image is still heavily defocused. Figure 5f shows the RD imaging result after phase error correction based on the proposed minimum entropy algorithm. It is clear that RD imaging results are defocused. Figure 5g shows the target high-resolution ISAR image obtained using the algorithm proposed in this paper. It can be seen from the figure that the imaging effect of the proposed algorithm is very good, which can effectively compensate for the phase error, and achieve high-resolution imaging of high-speed maneuvering targets.

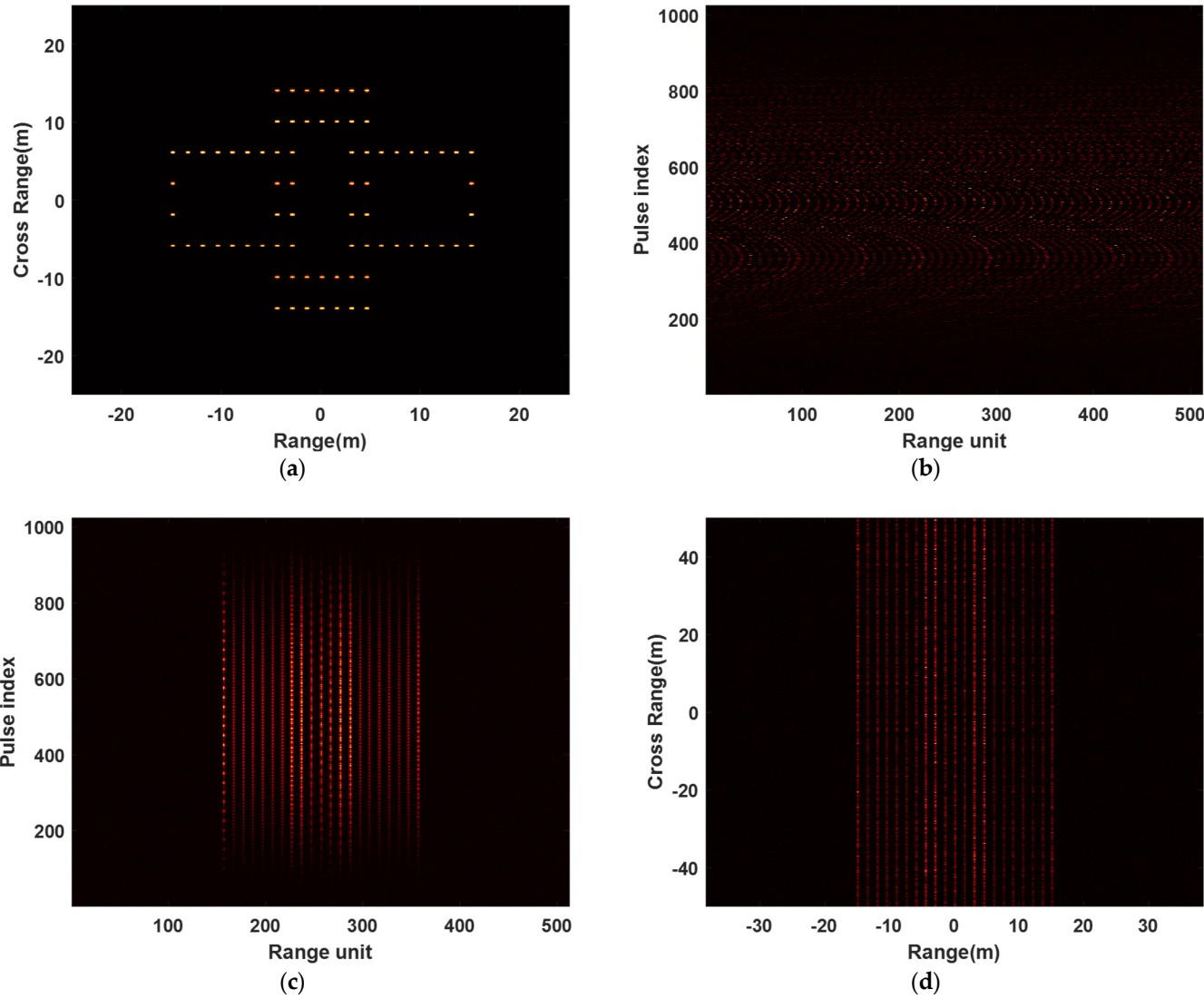

**Figure 5.** *Cont.*

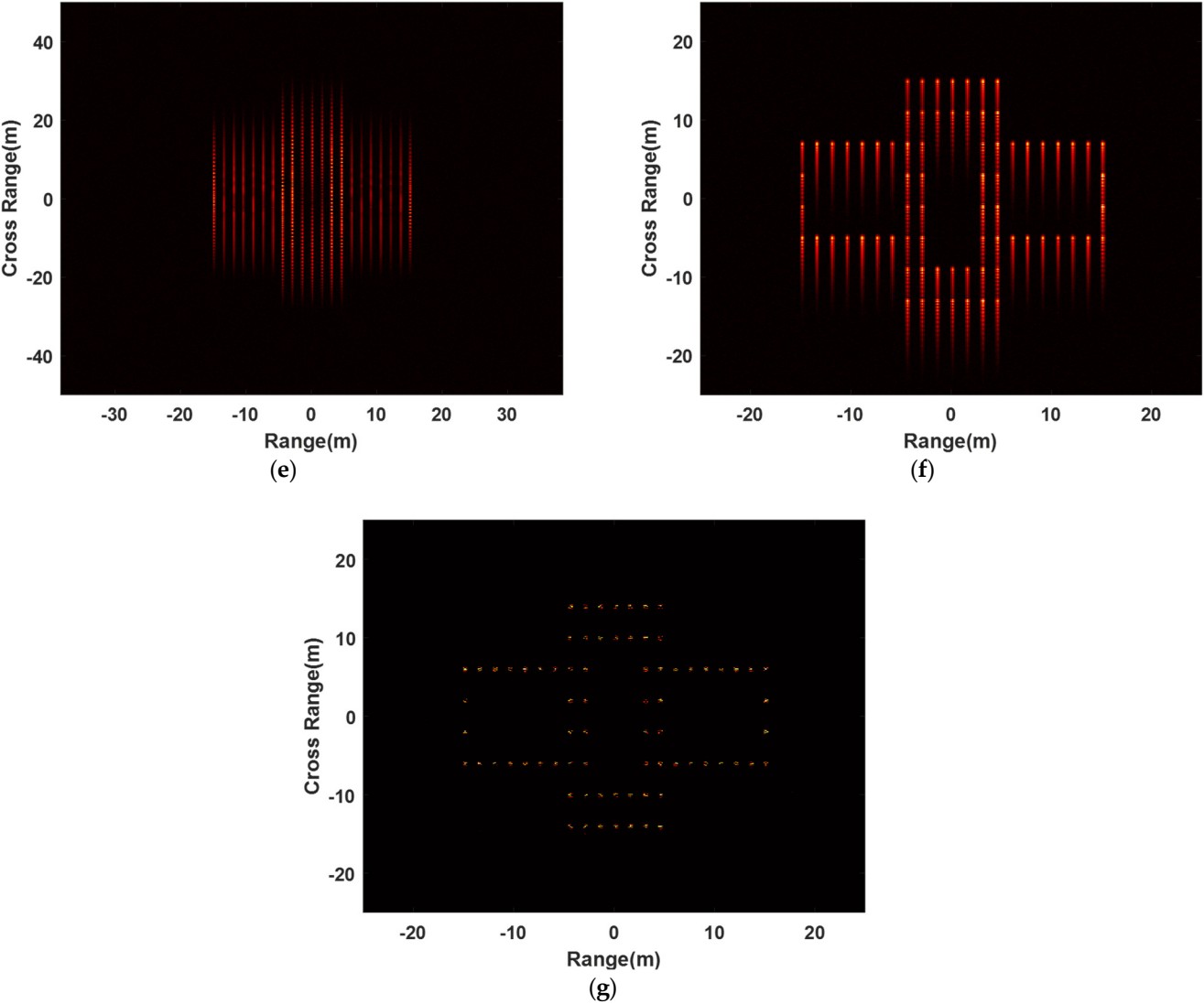

**Figure 5.** One-dimensional-range image and two-dimensional imaging results. (**a**) Ideal satellite imaging result. (**b**) One-dimensional-range image before envelope alignment. (**c**) One-dimensional-range image after envelope alignment. (**d**) RD imaging results before phase error correction. (**e**) RD imaging result after phase error correction based on rough estimation. (**f**) RD imaging result after phase error correction based on the proposed minimum entropy algorithm. (**g**) ISAR image obtained using the proposed algorithm.

In addition, we also give the RD imaging results based on rough estimation, RD imaging results after phase error correction based on the proposed minimum entropy algorithm, and the images of our proposed algorithm when the SNR is 5 dB and 0 dB in Figure 6. When the SNR is 5 dB, the imaging result obtained using FFT in the azimuth dimension after phase error correction based on rough estimates, the RD imaging result after phase error correction based on the proposed minimum entropy algorithm, and the image obtained using the proposed algorithm are shown in Figure 6a, Figure 6b, and Figure 6c, respectively. Figure 6d–f show the images when the SNR is 0 dB. It can be seen from the figure that the RD has basically failed when the phase error is corrected using only rough estimates, and the smaller the SNR, the more serious the defocusing of the image obtained using RD is. After using the minimum entropy correction error proposed in this paper, RD can obtain better imaging results; however, there is a defocusing phenomenon in the fast time dimension, which is because the estimated value of the target's high-order motion parameters, obtained using the algorithm, has a large gap with the real value,

and the precision of phase error compensation is poor. However, the proposed algorithm can obtain better imaging results even at a low SNR, and the position and shape outline information of the target can be obtained. However, when the SNR is low, the imaging results obtained using our proposed algorithm also have defocusing problems, which is because the accuracy of the estimation of error parameters is affected by the SNR. The lower the SNR is, the greater the difference between the estimated error parameters and the real value will be. When the phase error is not compensated for well, the image will defocus.

The average rough estimate values and the exact estimate values in the simulation experiment with different SNR of the above satellite model of 500 Monte Carlo experiments are shown in Table 3. From the table, we know that the rough estimate is close to the real value, but the parameter value is closer to the real value after accurate estimation, and the higher the SNR, the closer the estimated value of the parameter is to the real value. The parameter estimated using the combination of minimum entropy and SBL is closer to the real value than the parameter estimated using the direct minimum entropy algorithm. In addition, Table 3 also shows the image entropy of the ideal image, the image of RD, and the image of the proposed algorithm under different SNR. It can be seen from the table that when the SNR is 10 dB, the difference between the image entropy of the proposed algorithm and the ideal image is small. Therefore, the above analysis shows that the proposed algorithm has good accuracy and strong robustness.

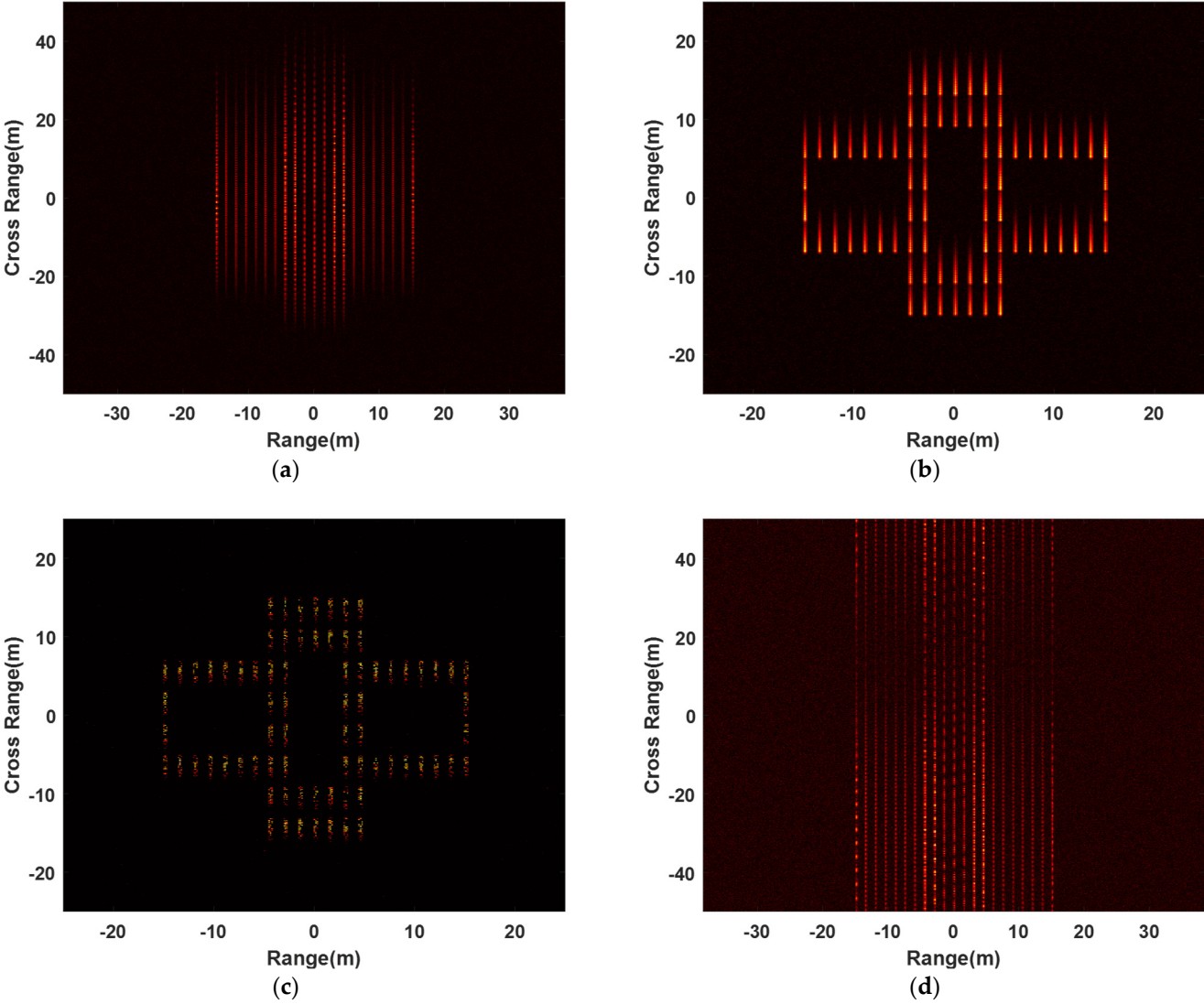

**Figure 6.** *Cont.*

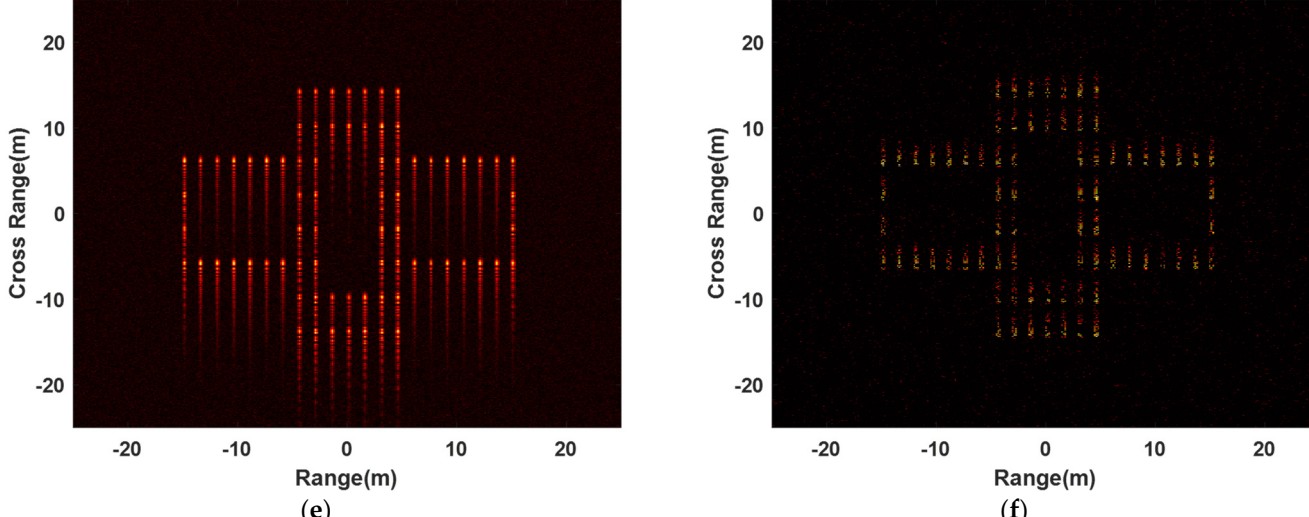

**Figure 6.** The imaging results of RD and the proposed algorithm with different SNR. (**a**) RD imaging result after phase error correction based on rough estimation when SNR is 5 dB. (**b**) RD imaging result after phase error correction based on the proposed minimum entropy algorithm when SNR is 5 dB. (**c**) ISAR image obtained using the proposed algorithm when SNR is 5 dB. (**d**) RD imaging result after phase error correction based on rough estimation when SNR is 0 dB. (**e**) RD imaging result after phase error correction based on the proposed minimum entropy algorithm when SNR is 0 dB. (**f**) ISAR image obtained using the proposed algorithm when SNR is 0 dB.

**Table 3.** Rough estimate values and exact estimate values obtained using the proposed algorithm with different SNR.

| | SNR (dB) | 10 | 5 | 0 |
|---|---|---|---|---|
| **Estimate of target motion parameters** | Rough estimate value [velocity (KM/s), acceleration (M/s$^2$), acceleration rate (M/s$^3$)] | [10, 120.11, 40.01] | [10, 119.81, 40.04] | [10.001, 120.41, 39.93] |
| | Exact estimate value in experiments of RD image based on the proposed minimum entropy algorithm [velocity (KM/s), acceleration (M/s$^2$), acceleration rate (M/s$^3$)] | [10, 120.029, 39.971] | [10, 119.977, 40.015] | [10, 120.031, 39.911] |
| | Exact estimate value in experiments of the proposed image algorithm [velocity (KM/s), acceleration (M/s$^2$), acceleration rate (M/s$^3$)] | [10, 120.001, 40.001] | [10, 119.958, 40.005] | [10, 120.011, 39.985] |
| **Image entropy** | Ideal image | −1.6048 | −1.6061 | −1.6112 |
| | RD image based on rough estimate | −1.6396 | −1.6506 | −1.6587 |
| | RD image based on the proposed minimum entropy algorithm | −1.6124 | −1.6327 | −1.6451 |
| | Image obtained by the proposed algorithm | −1.6070 | −1.6254 | −1.6320 |

For the measured data, in order to make phase error correction more accurately, it is often necessary to select the order of the maneuvering target echo polynomial before error correction. As shown in Figure 7, we used the Yak-42 aircraft to conduct actual simulation. For the system for obtaining Yak-42 aircraft echo data, the bandwidth and pulse repetition frequency of the radar employed to collect ISAR echo data are 400-MHz and 300-Hz, respectively. Both the range dimension sampling number and the azimuth

dimension sampling number are 256. Figure 7a,b are the imaging results of aircraft echo modeling into second-order and third-order polynomials using the proposed algorithm, respectively. Their image entropy values are 1.55453 and 1.5341, respectively, and have been indicated on the images. By comparing the two images, it can be seen that the tail of the aircraft in Figure 7b has a better focusing effect and a lower image entropy. Therefore, the third-order polynomial signal is more fitting to the real echo signal of the aircraft.

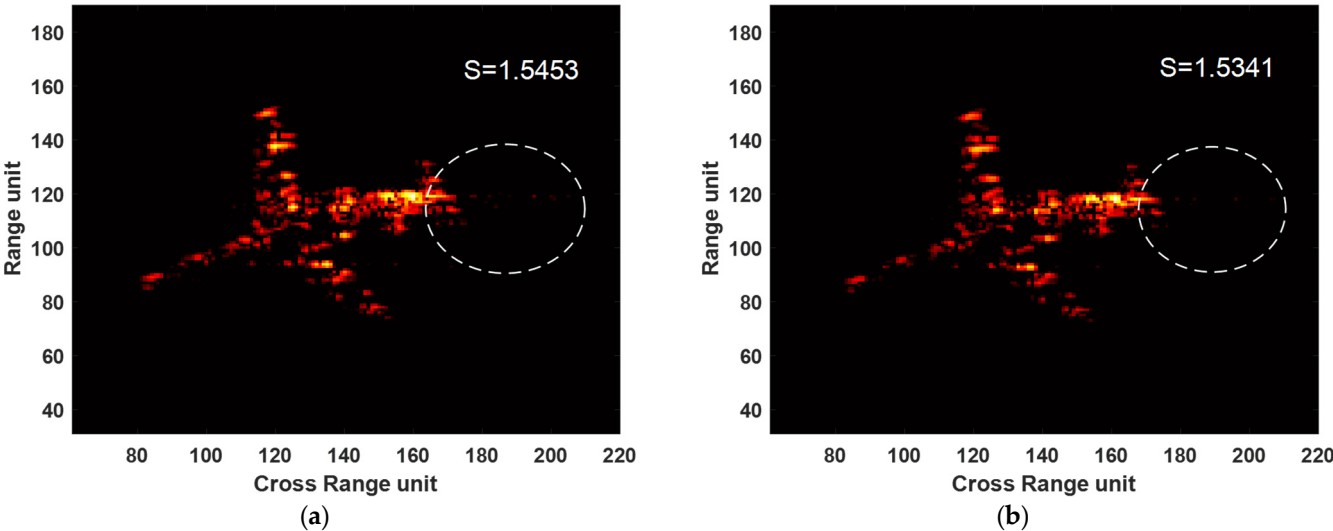

**Figure 7.** The measured data imaging results obtained using the proposed algorithm corresponding to different polynomial orders. (**a**) Imaging results of aircraft echo modeling into second-order. (**b**) Imaging results of aircraft echo modeling into third-order.

In fact, for the maneuvering target, the higher the order of distance polynomial, the more accurate the description of the real maneuvering characteristics of the target; however, the operational complexity will also increase. Therefore, the modeling of the polynomial order should choose the appropriate order according to the actual situation. In practice, it is generally possible to describe the motion characteristics of the target more accurately by modeling second-order or third-order polynomials, while the accuracy of the higher-order polynomial model is not very high, and the more parameters are estimated, the larger the calculation amount will be.

## 5. Conclusions

In this paper, we combine the parameter estimation method based on minimum entropy constraint with fast SBL to propose a high-speed maneuvering target high-resolution imaging method. Compared with some existing fast SBL algorithms, the proposed fast SBL algorithm does not use any approximation method, but uses the LC decomposition algorithm to represent the inverse matrix. Based on the decomposition expression of the inverse matrix, the operation involving the inverse matrix can be solved using FFT/IFFT. The theory and experiment prove that the proposed algorithm can achieve high-resolution ISAR imaging efficiently and accurately. In addition, the fast SBL algorithm based on the LC decomposition method has a lower computational complexity than the fast SBL algorithm based on GS decomposition. The estimation of motion parameters is also a key step in the imaging of high-speed moving targets. The accuracy of parameter estimation greatly affects the imaging results. The minimum Tsallis entropy is adopted for parameter estimation in this paper, which takes into account the imaging quality of the whole scene. Therefore, it has a stronger robustness, can achieve global optimization, and can obtain more accurate parameter values.

**Author Contributions:** Conceptualization, S.X.; data curation, Y.W.; investigation, Y.W. and J.Z.; methodology, S.X. and Y.W.; project administration, F.D.; software Y.W.; visualization Y.W.; supervision, J.Z.; resources, J.Z.; writing—original draft, Y.W.; writing—review and editing, S.X. and F.D. All authors have read and agreed to the published version of the manuscript.

**Funding:** This work was supported by the National Natural Science Foundation of China under grant no. 62271363.

**Data Availability Statement:** Not applicable.

**Conflicts of Interest:** The authors declare no conflict of interest.

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
