# Peer review of "High-Speed Maneuvering Target Inverse Synthetic Aperture Radar Imaging and Motion Parameter Estimation Based on Fast Spare Bayesian Learning and Minimum Entropy"

_remotesensing, doi:10.3390/rs15133376_

Round 1

Reviewer 1 Report

The authors proposed an interesting imaging algorithm which combines the SBL and the minimum entropy method, and the SBL is also improved to be more efficient. However, there are some questions unclearly dealt with.

1) The fast SBL and the entropy method using quasi-Newton, are clearly addressed respectively, but their combination is very obscure.

2) The notations in the equations are improperly displayed and the consistency of the notations throughout the paper are not coherent.

3) Comparison between the proposed method with the RD with high-order(3rd phase) error correction is not enough.

4) SBL, as a parameterized algorithm, require, in general, higher SNR than the FFT-based method such as RD. But the authors claim it needs lower SNR.

5) Are the aircraft data real, simulated or semi-physically simulated?

There are a few grammar mistakes which make the manuscript difficult to understand.

Reviewer 2 Report

The manuscript presents a high-resolution imaging method for high-speed maneuvering targets, which uses the fast-sparse Bayesian learning (SBL) algorithm and the minimum entropy algorithm for ISAR high resolution imaging and motion parameter estimation, respectively. Simulation results show that that the algorithm is effective. But In my opinion, it can be considered for publication only after a Major Revision. The following issues and hesitations should be addressed:

1. Please check that the variables in all formulas are defined, such as I, δ_i and λ in (2).

2. Why is the motion error in (1) defined as a polynomial of order 3, not order 4, order 5, etc.

3. The article mentions that RD algorithm will fail in the presence of high-order motion errors. If RD algorithm is combined with high order motion error compensation, such as the minimum entropy algorithm, is it possible to achieve ISAR imaging of high-speed maneuvering targets? If not, please explain why

4. Please give a detailed description of each picture, as shown in Figure 5 and Figure 6

5. How do the rough and exact estimate value in Table 6 come from?

6. In the results of measured data, please describe in detail the system for obtaining echo data. 

7. This paper gives a comparison of the compensation effect of the second order motion error and the third order motion error, is the third order the best? Do you get better images if you model with higher order motion errors? If RD algorithm is used to image the estimated 2th-order motion error and 3th-order motion error, what is the image effect?

Minor editing of English language required

Reviewer 3 Report

The title suggests high speed manuvering target imaging and attempt to recover sparse target model based on SBL. 

There are several points that are not clear in the context

SBL is described for signals mixed with noise, which is a critical performance measure for sparse target recovery. The simulation target model consists of a number of points with no background noise. While the theoretical decription involves gaussian noise and target model, the simulation model does not fully reflect these performance measures. It is odd that  0 dB SNR signal produce ISAR images.

It appears that a contant translational velocity is taken in simulations, which is a very simple ISAR imaging case. A simple signal analysis may provide a good estimation without complicated iterations. 

The simulation parameters are not adequate. Normally, satellites operate above 600km and 96km distance is too short. In this attitude, the translational distance is below 10km/s, while 50km/s is taken in this paper.

I wonder what is the definition of SNR here. It looks like the SNR is measured in the image domain, not in the raw signal. Regardless, the definition should be made clearly. In this case, SNR 0 should not produce any meaningful outcome and the iteration should have failed. 

Experimental test is made upon flying aircraft, which corresponds to the application of the proposed algorithm. However, I don't see how the proposed algorithm is applied to produce the ISAR imaging.

ISAR imaging is well known technique and high speed manuevering target images is not easily obtained without accurate velocity estimation. There are numerous works on this problems, and I don't really see whether this pape provides approprate design model and sufficient analysis. 

There are some cases that can be improved but overall, I dont see any problem.

Reviewer 4 Report

This study provides a framework for high resolution imaging of high speed maneuvering targets. The proposed algorithm has two main parts. For high resolution imaging the authors propose a fast SBL algorithm that can reduce the computational cost of SBL without reducing the accuracy. The second part of the proposed algorithm includes transforming the motion parameter estimation into a minimum entropy optimization problem which is then solved using quasi-Newton algorithm that can effectively estimate the parameters. The results indicate that the proposed framework is fast and accurate compared to traditional methods.

Comments

1) There is no line number which makes it impossible to point out to specific lines. In abstract, sparce Bayesian Learning should start with Capital letters. The font used for the word “which” is different from the rest of the text. In keywords there is reductant space before one of key words. The document should be extensively checked for formatting issues. 

2) Many formulas and symbols are not visible or readable. 

3) The English should be checked and improved throughout the introduction. There are a number of very long and hard to follow sentences. Some sentences and phrases are grammatically correct, but the language used seems informal and can be improved. An example is the first sentence of the introduction: “… so it has been paid more and more attention and development in the field of remote sensing detection”. I highly recommend that the authors make significant improvements to the English language used in the introduction.

4) In the first paragraph of the introduction a reference should be mentioned for the last statements.

5) In the second paragraph of the introduction, the authors state that “It is well known that the electromagnetic scattering characteristic of radar target is in the high frequency region, and the echo signal of the target can be characterized by a few important scattering centers”. They should provide a reference.

6) In the third paragraph of introduction, the authors state “SBL is a very important optimization algorithm”. Why is it important and for what purpose?  I believe the authors mean it’s among the most widely used algorithms. I suggest the author rewrite this.

7) In third paragraph of introduction, This sentence should be more clearly written : “The optimal mean value of the posterior distribution obtained by iteration is the reconstructed signal value finally obtained.

8) At the end of the fourth paragraph of introduction, the authors state that they will introduce a new SBL algorithm. Then start the next paragraphs with new topics. I strongly recommend the authors restructure the introduction. The current version looks like two separate introductions put one after another.  

9) In page 5, third paragraph, the authors use the phrase “SBL has been widely concerned”. I suggest the authors rewrite this in a more common and formal style of English. Also, in fourth paragraph authors use the term “detailedly” which is not a common word. There are other examples of this in the manuscript. I strongly suggest the authors review and revise the manuscript.

10) In section 4.1 and 4.2, (for example in the first and second paragraph of section 4.2) there are some information about the methodology. I strongly suggest the authors add a section for experiment design and put all such information in that section. The results section should focus on results and discussion rather than experiment design.

11) The caption of Figures 3, 5 and 6 should include information about each subfigure individually.

12) Table caption numbers should be changed to English numbers. The information presented in Table 3 is not clear. It should be modified.  

I suggest the authors improve the English in introduction. The rest of the manuscript seems fine.

Round 2

Reviewer 1 Report

Difficult to find where it‘s revised in the manuscript.

The notation and the quality throughout the manuscript be doublechecked.

Author Response

Thank you for your suggestion. The revision have been marked in yellow in the resubmitted article. The English usage of the paper has been improved, and some grammatical and spelling errors has been corrected and unclear parts in the original version has been rewritten.

Reviewer 2 Report

Accept in present form

Author Response

Thank you for your suggestion.